# Migraine and Sleep—An Unexplained Association?

**DOI:** 10.3390/ijms22115539

**Published:** 2021-05-24

**Authors:** Marta Waliszewska-Prosół, Marta Nowakowska-Kotas, Justyna Chojdak-Łukasiewicz, Sławomir Budrewicz

**Affiliations:** Department of Neurology, Wroclaw Medical University, 50-556 Wroclaw, Poland; marta.nowakowska-kotas@umed.wroc.pl (M.N.-K.); justyna.ch.lukasiewicz@gmail.com (J.C.-Ł.); slawomir.budrewicz@umed.wroc.pl (S.B.)

**Keywords:** migraine, sleep, sleep disorders, serotonin, melatonin, orexins, dopamine

## Abstract

Migraine and sleep disorders are common chronic diseases in the general population, with significant negative social and economic impacts. The association between both of these phenomena has been observed by clinicians for years and is confirmed by many epidemiological studies. Despite this, the nature of this relationship is still not fully understood. In recent years, there has been rapid progress in understanding the common anatomical structures of and pathogenetic mechanism between sleep and migraine. Based on a literature review, the authors present the current view on this topic as well as ongoing research in this field, with reference to the key points of the biochemical and neurophysiological processes responsible for both these disorders. In the future, a better understanding of these mechanisms will significantly expand the range of treatment options.

## 1. Introduction

Migraine is a widespread neurological disorder, affecting approximately 1 billion people worldwide. According to the Global Burden of Disease Study 2016, migraine is the second most common cause of disability and is the second most prevalent neurological disease. Its estimated 1 year prevalence is approximately 15% in the general population, three times more common in women than men [1,2]. The peak of prevalence affects people between 35 and 39 years, about 75% of migraine headaches start before the age of 35 years [2,3].

The term “migraine” is derived from the Latin word “hemicrania”, which means half (hemi) skull (crania). The disorder is diagnosed based on criteria defined by the International Classification of Headache Disorders (ICHD-3) (3rd Edition, 2018) [4]. According to the definition, migraine involves a unilateral or bilateral, pulsating headache aggravated by physical activity. The pain may be accompanied by nausea, vomiting, and photo- and phonophobia. In most patients the intensity of pain is from moderate to severe during attacks [2,4,5]. The ICHD-3 criteria divide migraine for three main categories: migraine with aura, migraine without aura and chronic migraine [4]. Migraine with aura is characterized by transient focal neurological symptoms (visual, sensory, speech, language, motor, brainstem or retinal) which usually precede or sometimes accompany the headache. The most common aura is the visual aura occurring in 90% of migraine patients with aura [2,5].

Sleep is a spontaneous and periodic physiological state consisting in the absence of motor activity, decreased reactivity to stimuli and a stereotypical position. Behavioral changes during sleep are accompanied by a reorganization of brain activity. Sleep is one of the most important physiological processes and is considered to be the rest and convalescence phase essential in preparing the body for subsequent wakefulness [6,7]. Sleep also supports cognitive functions, mood, and memory, and affects the proper functioning of the endocrine and immune systems [7,8]. Rechtschaffen’s experimental studies on sleep deprivation in rats showed that complete sleep deprivation led to the death of all test animals within 2–3 weeks [9].

Sleep disorders are classified according to several major classifications systems, such as the International Classification for Sleep Disorders, 3rd Edition (ICSD-3), the Diagnostic and Statistical Manual of Mental Disorders, 5th Edition (DSM-5), and the International Classification of Diseases, 10th Edition, Clinical Modification (ICD-10-CM) [10,11]. According to ICD-10-CM classification system, seven main categories of sleep disturbances are distinguished. The main category of sleep problems includes insomnias, parasomnias, hypersomnias, sleep-related breathing disorders, circadian-rhythm disorders, sleep-related disorders and other sleep disorders (Table 1) [10,11,12].

The earliest published research from the 19th century by Wright suggested an association between sleep and the occurrence of migraine [13,14]. In 1853 Romberg indicated that an attack of migraine can be stopped by sleep, the same observation was presented by Leveing in his publication “The effect of sleep on headache relief” in 1873 [15]. Up to half of headache patients appear to report various sleep disturbances and a significant relationship between headache and sleep is found when approximately 75% of cases of pain occur during sleep or immediately after waking up [16,17,18]. In most cases, sleep is a reliever of migraine attacks [19,20]. In a study based on an arctic population, sleep was the main protective factor against migraine attack. In this group of patients, episodic morning headaches were connected with lack of sleep or insomnia [21]. So, poor quality or duration of sleep could also be a trigger of migraine attack [22]. Numerous studies have shown that people with migraines have poorer quality of sleep than those who do not suffer from migraines, and that migraine prophylactic treatment significantly improves sleep quality [16,23,24]. People suffering from migraines are significantly more likely to suffer from poor sleep quality, insomnia and night-time fatigue. Attention has also been paid to the more frequent occurrence of migraine attacks in the morning, and almost half of these occur immediately after waking up [25,26]. It is known that in patients with migraine there is a very common change in sleep pattern–in this case, the percentage of stage III and IV NREM sleep before an attack increases and the percentage of REM sleep increases when the patient wakes up with a migraine [27].

In the condition which we called “weekend migraine” headache occurs after oversleeping. Park et al. suggested that exposure to excessive sleep and sleep deprivation appear to the most frequent causes of migraine attacks in the morning [28]. Moreover, some patients with headache triggered or relieved by sleep demonstrate some behaviors, which predispose them to deepening sleep disturbances [29]. Sleep problems such as sleep deprivation, oversleeping, or lack of regularity could be an important factor in the transition from an episodic to chronic form of migraine [19,25,30].

The connection between sleep disturbances and migraine is very well documented but the pathogenetic relationship between sleep and migraine is unknown. Scientists have tried to describe these common interactions based on common anatomical areas of the brain such as the hypothalamus and the supraoptic nucleus. In recent years, there has been increased interest in finding common biochemical mediators, e.g., serotonin, melatonin, orexins, dopamine and adenosine, involved in both migraine and sleep disorders. This article presents the current state of knowledge regarding the neuroanatomical and biochemical links between sleep disorders and migraine.

## 2. Methods

The authors conducted a literature search focused on the topic of the relationship between sleep and migraine. The key search terms applied in PubMed via MEDLINE and Google Scholar were “sleep” or “sleep disorders” and “migraine” or “headache”. The online search covered a publication period from 2010 until 31 January 2021. As a result, 916 records were identified and screened. From those approximately 130 abstracts were found to be relevant to the subject; duplicate publications were removed. Selected publications providing basic knowledge and historical research come from earlier years. The reference lists from eligible publications were searched for their relevance to the topic. Reviews and research studies, classified according to their relevance, were initially included, with subsequent exclusion of conference abstracts and papers written in languages other than English. The studies not unpublished and not based on current criteria for the diagnosis of migraine and sleep disorders were excluded.

## 3. Pathogenesis of Migraine and Sleep Disorders

### 3.1. Migraine

The pathogenesis of migraine is unknown. It is most likely a polygenetic channelopathy, which is characterized by a predisposition to increased vasomotor reactivity on the basis of seizure changes in the central nervous system [2,5]. Two pathogenetic theories are currently being discussed: vascular and neuronal. Additionally, it has been documented that a third system influencing the development of a migraine attack are the cortex centers [5,31].

The theory of the functional trigeminovascular system has been developed, which includes the brain, brain vessels, and trigeminal nerve with nuclei in the brainstem [2,31]. This system is activated during a migraine attack, causing the release of a number of neuropeptides from the sensory endings, such as the calcitonin gene-related peptide (CGRP), substance P, a pituitary adenylate cyclase-activating polypeptide (PACAP38), vasoactive intestinal peptide (VIP), neurokinin A, and nitric oxide synthase (NOS) [31,32]. CGRP is a common neuropeptide of the sensory system, which acts via vasodilation through the type 1 receptor. Its expression has been revealed in numerous areas of the central and peripheral nervous system: in the nucleus and trigeminal nerve, the hippocampus, the amygdala, pons, and paraviductal grey matter [32,33,34].

The released neuropeptides lead to the onset of neurogenic inflammation (swelling around the arachnoid vessels), vasodilation, and increased cerebral blood flow. Changes in the flow in the central nervous system are associated with the occurrence of cortical spreading depression (CSD), which begins in the occipital and parieto-occipital regions, then moves through the cerebral cortex and stops at the medial and lateral sulcus [34]. At the same time, histamine is released from mast cells, and the aggregation of serotonin-releasing platelets occurs in capillary vessels. The stimulation of the nuclei of the trigeminal nerve provokes pain [31,34,35].

The biochemical theory of migraine development is primarily related to the metabolism of serotonin (5-hydroxytryptamine, 5-HT), a biogenic amine that acts as one of the neurotransmitters within the CNS [36]. Serotonin is stored in the nuclei of the raphe stretching from the midbrain to the medulla. With the help of neural connections, these connect to the cortex of the brain, the limbic system and the subcortical nuclei. Serotonin plays a role in keeping artery walls taut; it also narrows arteriovenous connections [37,38].

Serotonin affects the cerebral cortex through 5HT1 and 5HT2 receptors, the vascular system and the nucleus of the trigeminal nerve with 5HT1D alpha and beta and 5HT 2B/C, and through 5HT1D and 5HT3 on the pain system [38,39].

A relationship has been demonstrated between the contraction of carotid vessels and 5HT1D receptor stimulation. The 5HT3 receptor plays a role in the development of nausea and vomiting during a headache attack. The level of 5-HT remains under self-control, which is the responsibility of the 5HT1B/1D receptor, which inhibits the secretion of this amine [37,38]. A reduction in its concentration in the blood serum causes the arterial wall to relax, with a simultaneous increase in its permeability to substances that lower the pain threshold for nocturnal receptors. The pathomechanism presented above enables combination of the vascular biochemical theories [36]. There are many possible causes in the literature for abnormal 5-HT metabolism. These include disturbances in the synthesis of the amine itself leading to the formation of unstable forms, an incorrect disintegration mechanism or receptor dysfunctions [32,37].

Disruptions in the dopaminergic system are also taken into account when considering the pathogenesis of migraine. Dopamine is a biogenic amine derived from the transformation of L-tyrosine, as well as being the starting substance for the substances norepinephrine and adrenaline. It acts on the cell through specific metabotropic receptors located in the pre- and postsynaptic membranes [40]. These receptors form two families consisting of the D1 and D5-type receptors (acting through the Gs proteins and stimulating cAMP synthesis), and a group consisting of the D2, D3 and D4 receptors (acting through the Gi proteins, reducing cAMP turnover and inhibiting the turnover of cAMP). CNS dopaminergic neuron populations have been classified and labelled as A8-16 cell groups [32,41]. Research suggests the presence of postsynaptic dopaminergic and serotonergic receptor overactivity, as evidenced by the presence of symptoms such as yawning, eating disorders, nausea and vomiting before an attack. D2 receptor antagonists have been shown to prevent the development of migraine during the prodromal phase. D1 and D2 receptors are located in the nucleus and trigeminal nerve, and D3 and D4 receptors on lymphocytes and in the peripheral blood of patients with migraine [42,43].

Recently, the role of nitric oxide (NO) in the pathomechanism of migraine headache has been increasing [44]. It has the characteristics of a free radical, is released from the vascular endothelium, and its action is associated with the relaxation of vascular smooth muscles. Currently, it is associated with the activation of the trigeminovascular system, the release of histamine from the perivascular mast cells located in the meninges, as well as the effect of neuropeptides on the vascular endothelium and cortical ischemia. Its role as a transmitter in short- and long-term electrical activation, in pain processes and in the control of the sympathetic nervous system has been suggested [30,31,45,46].

In recent years, a “central hypothesis” has emerged that takes into account the involvement of the sympathetic and parasympathetic systems in the pathogenesis of migraine. The main neuromediators of the former are norepinephrine and neuropeptide Y. Released at the ends of sympathetic fibers, these neuromediators cause narrowing of the CNS vessels. Neurotransmitters associated with the parasympathetic system include: acetylcholine, vasoactive intestinal peptide, and nitric oxide synthetase. The parasympathetic system in the CNS has a vasodilating effect [47,48,49]. There are increasingly frequent reports of disorders of the autonomic system in patients with migraine [50]. These reflect the clinical symptoms observed in patients (vasomotor and heart rhythm disturbances, nausea, vomiting and diarrhea) during the headache attack itself. Sympathetic insufficiency is suggested both in the interictal period and in the course of pain. In both described groups of patients, these periods exhibit low levels of norepinephrine. Some authors claim that in this disease there is instability of the autonomic system with progressive hypothyroidism, and then secondary hypersensitivity [51,52].

Evidence is still being sought concerning the possible effects of melatonin and orexins on the development of migraine. Proving their role in the pathogenesis of migraine headaches could expand the available drug arsenal. Nevertheless, despite the possible potential benefits of melatonin, further observations are required [53]. In research on new anti-migraine drugs, some hopes have been placed on the rexants, which are orexin receptor antagonists. Their possible therapeutic effect has been seen in the relationship between migraine and sleep, and in the hypothesis that drugs that potentially improve the quality of sleep could contribute to the effective prevention of migraine attacks [54]. However, clinical trials have not confirmed the efficacy of filorexant or ramelteon in reducing migraine attacks [55,56].

### 3.2. Sleep Disorders

Sleep can be broadly segmented into rapid eye movement (REM) sleep and non-REM (NREM) sleep. There are two phases of REM sleep: phasic and tonic. Phasic REM sleep contains bursts of rapid eye movements, respiratory variability, and brief electromyography activity (occasionally seen as muscle twitches). More limited motor activity occurs during tonic REM sleep, with few eye movements [6,8].

Constantin von Economo used his clinical observations from 1923 to 1925 to propose that the “sleep-regulatory center” (Schlafsteuerungszentrum) is located near the oculomotor nucleus, the aqueduct of the third ventricle and the infundibular region [57]. Later, sleep cycle observation led to the development of the hypothesis of reciprocal discharge by two brainstem neuronal groups [58]. The neurobiological processes that regulate sleep are very complex. The most important neurotransmitter systems are presented in Table 2.

The first sleep phase is NREM, and the subcortical sleep promoting systems are mostly GABAergic and spread throughout the brain. Ventrolateral preoptic neurons in the hypothalamus (VLPO) consist of neurons active during sleep. They can be divided into two subgroups of neurons: tightly clustered neurons which project to the tuberomammillary nuclei promoting NREM sleep; and diffusely distributed neurons (also called extended VLPO–eVLPO) which project to locus coeruleus (LC), dorsal raphe nuclei (DRN) and to the interneurons of the lateral dorsal tegmental (LDT) and pedunculopontine tegmental (PPT) region, participating in the control of REM sleep [59]. 80% of VLPO neurons are GABAergic and galaninergic in nature [60].

Sleep-promoting neurons of VLPO are activated by sleep-inducing factors including adenosine, prostaglandin D2 and warmth, thus heating this area of the brain increases their activity and decreases wake state [61,62]. Arousal systems inhibit the activity of GABAergic and galaninergic VLPO neurons. These are inhibited both by noradrenaline through activation of postsynaptic α2-adrenoceptors [63] and also by acetylcholine [64]. The second important NREM controlling system is the parafacial zone (PZ) consisting of GABAergic neurons, which are necessary for the induction of deep NREM, expressed as slow-wave sleep [65]. The ventrolateral tegmental nucleus (VTA), which was first discovered to be a part of the arousal system because of its dopaminergic neurons, has been proven to play a role in promotion of NREM sleep because its GABAergic neurons inhibit both dopaminergic neurons and also the lateral hypothalamus region [66,67] Other GABAergic neurons located in the ventrolateral periaqueductal grey region (vlPAG) stabilize NREM sleep, reducing REM sleep episodes at the same time [68]. Neurons expressing the adenosine receptor A2a located in the nucleus accumbens and in rostral striatum also promote NREM sleep by inhibition of pallidum [69]. Glutaminergic neurons located in two regions promote NREM sleep: deep mesencephalic nucleus (DpMe) and perioculomotor neurons [70,71]. NREM sleep-active neurons located in the median preoptic nucleus (MnPO) fire faster during prolonged wakefulness (which increases sleep pressure) [72]. Together with VLPO neurons, the MnPO neurons strongly inhibit arousal-promoting brain regions, including the cholinergic neurons of the basal forebrain, orexin neurons in the lateral hypothalamus, tuberomammillary nucleus (TMN), DRN, median raphe, parabrachial nucleus (PB) and LC [73].

Control of REM sleep is based on neurons mostly located in the brainstem and divided into REM-on and REM-off neurons [74,75]. The main group of REM-on neurons are located in the sublaterodorsal nucleus (SLD); they are mainly glutaminergic in nature and project rostral to the forebrain and caudally to the medulla and spinal motoneurons [76]. The REM-off neurons consist of the cholinergic pedunculopontine tegmental (PPT) and lateral dorsal tegmental nuclei (LDT) neurons, aminergic locus coeruleus neurons (LC), serotoninergic dorsal raphe nuclei (DRN) and GABAergic neurons of vlPAG [77]. All these populations project to REM-on neurons and inhibit them. On the other side, the REM-off neurons are inhibited by a subpopulation of GABAergic neurons of SLD [75]. As an additional regulatory mechanism, the orexinergic (also referred to as hypocretin) neurons located in the lateral hypothalamus are involved. They fire maximally in the waking state, show occasional bursts during phasic REM sleep state, and are suppressed during NREM sleep, strongly activating REM-off populations of neurons located in PPT-LDT regions [78]. They cease activity during sleep, probably due to inhibitory input from GABAergic sleep-active neurons located in the basal forebrain and preoptic area [79].

The role of serotoninergic neurons in sleep control is currently a matter of debate. Initially, their role was thought to be purely promotion of wakefulness, but recent research indicates that they are also essential for initiating and maintaining sleep [80,81,82,83]. An explanation for this paradox is still being developed from animal studies. It is possible that it may be due to the significant variation in neuronal subpopulations [84].

Interference between sleep and migraine may occur on several levels. The first level is seen in a change in the concentration of certain neurotransmitters; the second is a change in cortical excitability; the third–energy levels; and the last–neurogenesis. The common anatomical structures of migraine and sleep disorders are shown in Table 3.

The concentration of the strong natural vasodilator adenosine increases with the duration of sleep deprivation and during migraine attacks and the precipitating effect of adenosine administration on migraine attacks has been stated [85]. One of the mechanisms underpinning this finding could be the overstimulation of A1 receptors, which may cause increased susceptibility to the formation of cortical spreading depression (CSD) [86]. Other proposed mechanisms of increased CSD susceptibility due to sleep deprivation are Ca2+ dependent pathways, an increase in the excitatory glutamate levels in the cerebral cortex, and increased expression of glutamate receptors in certain parts of the cortex [87,88,89].

The dysregulation of cyclic secretion of melatonin, an antinociceptive factor which operates via the arginine/NO/KATP pathway, can also occur in migraine patients as melatonin level decreases during an attack [90,91].

Lack of sleep may also influence cortex excitability by inhibition of the NA+/K+ TPase in the setting of the low energy level, a phenomenon exacerbated by decreased activity of anti-oxidative enzymes (i.e., superoxide dismutase) [92,93].

## 4. The Most Common Sleep Disorders in Migraine

In migraine patients, the most common sleep disturbances are insomnia, daytime sleepiness, sleep apnea and parasomnia [11,94]. Disturbed sleep in migraine patients has an impact on quality of life and is associated with increased disability [7,17].

### 4.1. Insomnia

The Most Common Sleep Problem in Patients with Chronic Migraine is Insomnia [29]. Insomnia is defined as difficulty with sleep initiation, duration and consolidation, despite adequate opportunity to sleep, with resulting daytime consequences [95,96]. Kim et al. showed a higher prevalence of insomnia in patients with migraine versus patients without headache (25.9% vs. 15.1%) [97]. Other studies [98,99] have also demonstrated that patients with migraine have an increased risk of insomnia. Insomnia impairs cognitive and physical functioning and is associated with a wide range of impaired daytime functions across a number of emotional, social, and physical domains. Patients with migraine, suffering with insomnia, have increased pain intensity, attack frequency and more probability of chronification of headaches [13,100]. The pathological mechanism underlying the relationship between migraine and insomnia remains unknown. It has been speculated that hypothalamus and brainstem dysfunction are common pathological mechanisms for both migraine development and insomnia. These structures are involved in sleep-wake physiology and pain modulation [8,16].

### 4.2. Obstructive Sleep Apnea

The term Obstructive Sleep Apnea (OSA) refers to a common disorder that causes patients to have temporary problems (apnea or hypopnea) with breathing during sleep. Sleep disturbances are associated with sleepiness, fatigue, insomnia, snoring, and subjective nocturnal respiratory disturbance. These breathing problems can awaken the person or prevent deep sleep. In patients with OSA, especially in cases of breathing difficulties, snoring or sleep apnea, morning headaches are common, but not migraine headaches [101,102]. In a study followed by clinical interview and polysomnography, Kristiansen and al. reported similar rates of sleep apnea among patients with migraine without aura, migraine with aura, and the general population [103,104].

### 4.3. Parasomnia

Other problems include parasomnias, which are a category of sleep abnormalities, defined as undesired behavior (e.g., sleepwalk, teeth-grinding) or unpleasant experiential phenomena (e.g., nightmares) during periods of sleep. Sleepwalking has been associated with migraine. Studies have shown a higher rate of somnambulism in patients with migraine with aura [105,106]. Somnambulism and migraine can appear at different ages, but especially in children, and could be linked with serotonergic metabolic dysfunction [107]. The most recent hypothesis linking parasomnia with migraine is dysfunction of the serotonergic pathway due to the well-known role of serotonin in both sleep-wake regulation and migraine pathogenesis [16].

### 4.4. Rest Leg Syndrome

The prevalence of rest leg syndrome (RLS) is 5–10% in the general population, but substantially higher in patients with migraine (8.7–39.0%) [108]. Similar to migraine, RLS predominates in women. In 2010, Chen found a correlation between RLS and migraine. Patients with migraine have more frequent symptoms of RLS than patients with tension headache. Additionally, patients with migraine and RLS have higher frequencies of photo- and phonophobia, vertigo, dizziness, tinnitus and neck pain [109]. Van Oosterhout et al. reported that prevalence of RLS in migraineurs is approximately 17% [110]. Didriksen showed that the prevalence of migraine with or without aura was higher among participants with RLS compared to participants without RLS. Moreover, in this study patients with frequent or severe symptoms of RLS had the highest risk of migraine and MA. The link between RLS and migraine is most likely the result of a shared dysfunction in the dopaminergic system in the A11 nucleus of the hypothalamus. Symptoms related to dopaminergic dysfunction in the prodromal phase of migraine, and the hypersensitivity of migraine patients to dopaminergic agonists [111,112] suggest that dopamine may play a central role in the pathophysiology of migraine. Suzuki et al. reported that the presence of migraines is associated with sleep disturbances and depression in Parkinson’s disease (PD) patients, while overall headache and migraine severity reduced after PD onset [113]. The A11 dopaminergic nucleus modulates neuronal activity in the trigeminocervical complex that is linked with important areas of the pain, including the cortex, thalamus, and the brainstem [30,114]. The A11 nucleus sends direct inhibitory projections to sympathetic neurons and the motoneuronal site of the spinal cord. Therefore, dysfunction of the A11 dopaminergic nucleus possibly causes or worsens migraine and RLS [43,115].

### 4.5. Bruxism

Patients with migraine also report an increased frequency of nightmares, maybe this is due to a change in sleep architecture in migraineurs. Very common in patients with migraine, bruxism is associated with temporomandibular joint dysfunction. It is possible that bruxism and TMD trigger migraine headache by activation of the trigeminal nerve, or that patients with migraine are more susceptible to pain from TMD secondary to central sensitization [116,117].

### 4.6. Narcolepsy

Narcolepsy and migraine may co-exist, but the association is inconclusive. On the one hand, Dahmen et al. reported a significantly increased prevalence of migraine among narcolepsy patients [118]. Similar observations were reported by Suzuki et al., who noted that migraine was more common in patients with narcolepsy and idiopathic hypersomnia than in healthy controls [119]. However on the other hand, another multi-center case-control study with 96 narcoleptics patients did not any significantly higher prevalence of narcolepsy with migraine [120]. Narcolepsy patients have headaches but not only migraine. In the pediatric population, migraine is associated with the risk of development of narcolepsy [121]. In the migraine population, a higher frequency of the HLA-DR2 antigen is found.

New research suggests that the orexynergic system plays an important role in the relationship between narcolepsy and migraine. Orexins are two neuropeptides (orexin A and B) synthesized in the lateral hypothalamus and involved in the modulation of homeostasis including wakefulness, autonomic regulation, and hormone secretion [122,123]. Orexin peptides bind to two different receptors-OX_1_R and OX_2_R-and levels are higher during periods of wakefulness. They are known to strengthen the neural networks of the hypothalamus and the brainstem to stimulate wakefulness and the NREM phase of sleep. Dysfunction of the orexynergic system in the hypothalamus and loss of orexinergic neurons are responsible for the symptoms of narcolepsy. It is likely that they also modulate the trigeminovascular system, and as such they have aroused the interest of migraine researchers [16,30,123].

## 5. Conclusions

The link between migraine and sleep disorders is underlined by evidence from epidemiological studies, their similar clinical picture, and shared anatomical pathways. Although many studies have improved our understanding of the subject in recent years, more research is needed. Research into anatomical structures and neuropeptides appears to be crucial as it provides insight into the mechanisms underlying the relationship between migraine and sleep disorders, but may also be important for deepening our understanding of migraine pathology and developing new therapeutic approaches.

In the management of migraine patients, the diagnosis and treatment of comorbid sleep disorders should be considered, as improved sleep is expected to also reduce the frequency and severity of headaches. However, appropriate and optimal migraine abortive and prophylactic treatment can prevent secondary sleep disorders, which may consequently reduce the quality of life of migraineurs and increase the frequency of migraine attacks.

## Figures and Tables

**Table 1 ijms-22-05539-t001:** Classification of sleep disturbances according to The International Classification of Sleep Disorders, third edition (ICSD-3) [11].

Major Diagnostic Sections	Definition	Disorder
Insomnia	Difficulty initiating or maintain sleep, poor quality of sleep	Chronic insomnia disorder Short-term insomnia disorder Other insomnia disorder
Sleep-related breathing disorders *	Abnormal respiration during sleep characterized by intermittent partial or complete upper airway obstruction	OSA disorders: OSA, adult, OSA pediatric Central sleep apnea syndromes: Central sleep apnea with Cheyne-Stokes breathing Central sleep apnea due to a medical disorder without Cheyne-Stokes breathing Central sleep apnea due to high altitude periodic breathing Central sleep apnea due to a medication or substance Primary central sleep apnea Primary central sleep apnea of infancy Primary central sleep apnea of prematurity Treatment-emergent central sleep apnea Sleep-related hypoventilation disorders: Obesity hypoventilation syndrome Congenital central alveolar hypoventilation syndrome Late-onset central hypoventilation with hypothalamic dysfunction Idiopathic central alveolar hypoventilation Sleep-related hypoventilation due to a medication or substance Sleep-related hypoventilation due to a medical disorder Sleep-related hypoxemia disorder
Central Disorders of Hypersomnolence	Daytime sleepiness not associated with disturbed sleep or misaligned circadian rhythms	Narcolepsy type 1 * Narcolepsy type 2 * Idiopathic hypersomnia Kleine–Levin syndrome Hypersomnia due to a medical disorder * Hypersomnia due to a medication or substance Hypersomnia associated with a psychiatric disorder Insufficient sleep syndrome *
Circadian Rhythm Sleep-Wake Disorders *	Sleep disturbance due to misalignment between environment ant the individual’s sleep-wake cycle	Delayed sleep-wake phase disorder Advanced sleep-wake phase disorder Irregular sleep-wake rhythm disorder Non-24-h sleep-wake rhythm disorder Shift work disorder Jet lag disorder Circadian sleep-wake disorder not otherwise specified
Parasomnias	Undesirable movements, behaviors, perceptions or dreams, that occur during sleep or arousals from sleep without conscious awareness	NREM-related parasomnias: * Confusional arousals Sleepwalking Sleep terrors Sleep-related eating disorder REM-related parasomnias: * REM sleep behavior disorder Recurrent isolated sleep paralysis Nightmare disorder Other parasomnias: Exploding head syndrome Sleep-related hallucinations Sleep enuresis Parasomnia due to a medical disorder Parasomnia due to a medication or substance Parasomnia, unspecified
Sleep Related Movement Disorders *	Simple, stereotypic movements that disrupt sleep	Restless legs syndrome Periodic limb movement disorder Sleep-related leg cramps Sleep-related bruxism Sleep-related rhythmic movement disorder Benign sleep myoclonus of infancy Propriospinal myoclonus at sleep onset Sleep-related movement disorder due to a medical disorder Sleep-related movement disorder due to a medication or substance Sleep-related movement disorder, unspecified
Others sleep disorders	Sleep disorders that cannot be appropriately classified elsewhere	

Sleep disorders marked with * appear to be significantly associated with migraine.

**Table 2 ijms-22-05539-t002:** Major sleep-related neurotransmitter systems and common points with sleep disorders.

Neurotransmitter	Potential Common Mechanism for Sleep and Migraine
Adenosine	▪ endogenus somnogen released from neurons or glia cells ▪ NREM and REM sleep induced through action of A1 or A2A receptors in: - basal forebrain, - lateral hypothalamus, - tuberomammillary nucleus (TMN), - ventrolateral preoptic neurons in the hypothalamus (VLPO)
Cholinergic system	▪ pedunculopontine tegmental (PPT) and lateral dorsal tegmental nuclei (LDT) neurons–promotion of wakefulness ▪ basal forebrain–promotion of wakefulness
Dopaminergic system	ventrolateral tegmental nucleus (VTA) located in midbrain–consolidate wakefulness periaqueductal grey neurons–antinociception
GABA	▪ ventrolateral preoptic neurons in the hypothalamus (VLPO), ventrolateral tegmental nucleus (VTA)–promotion of NREM ▪ parafacial zone (PZ)–induction of deep NREM ▪ ventrolateral periaqueductal grey region (vlPAG)–stabilization of NREM sleep, reduction of REM sleep
Galanin	ventrolateral preoptic neurons in the hypothalamus (VLPO)–promotion of NREM
Histamine	tuberomammillary nucleus (TMN) located in posterior hypothalamus–promotion of wakefulness
Melanin	lateral hypothalamic area–promotion of REM sleep, promotion of NREM sleep in some conditions
Noradrenaline	locus coeruleus––inhibition of the ventrolateral preoptic neurons in the hypothalamus (VLPO)–promotion of wakefulness
Orexin (also referred to as hypocretin neurons)	lateral hypothalamic area–promotion of wakefulness
Serotonin	dorsal raphe nuclei–inhibition of REM and initiation of sleep but also mood regulation, food intake, temperature regulation

**Table 3 ijms-22-05539-t003:** Anatomical structures involved in sleep and migraine regulation (modification based of [16,30]).

Anatomical Structures	Sleep	Migraine
**Brainstem** ▪ periaqueductal gray (dopamine) ▪ dorsal raphe nucleus (serotonin) ▪ locus coerulus (norepinephrine)	- promotion of wakefulness - stabilization of the walking state - control of the transition to sleep	- pain transmission - pain modulation
**Hypothalamus** ▪ posterior hypothalamus (dopamine) ▪ lateral hypothalamus (orexin)	- promotion of wakefulness, - regulation of circadian rhythm, - control of sleep-wake transition	- processing, transmission, modulation of pain
**Thalamus**	- promotion of wakefulness and integration of sub-cortical sleep-wake inputs	- processing and transmission of nociceptive information
**Cortex**	- promotion of wakefulness	- pain processing - pain modulation

## Data Availability

The data presented in this study are available upon request from the corresponding author. The data are not publicly available.

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
