# Peer review of "Migraine and Sleep—An Unexplained Association?"

_ijms, 2021, doi:10.3390/ijms22115539_

Round 1
Reviewer 1 Report
It explains the relationship between migraine and sleep in an easy-to-understand manner, and I think it is a worthwhile paper to publish.
It is known that sleep disorders become more prominent in Parkinson's disease patients than in normal patients as the symptoms progress. Suzuki et al. reported that the presence of migraines is associated with sleep disturbances and depression in PD patients, while overall headache and migraine severity reduced after PD onset (https://doi.org/10.1177/0333102417739302). It is one of the valuable papers reporting the association between migraine and sleep disorders and is recommended to be cited.
Author Response
Dear Reviewer,
Thank you very much for your review. All comments and suggestions from the reviewers have been included in the manuscript. English language and style was check by native speaker.
Hopefully, the revised version of the manuscript, considering the above issues, would be found suitable for publication.
Authors
Reviewer 2 Report
This is a nice review paper. The aim of the paper is delineated. Methods are presented with key terms and the research period as well.
In addition, the paper gives comprehensive insight into the physiology and pathophysiology of sleep regulation, migraine and potential interactions between sleep and migraine. Limitations and the need of further research are discussed.
Please accept in present form.
Author Response
Dear Reviewer,
Thank you very much for your nice and review.
Authors
Reviewer 3 Report
The authors conducted a review on the comorbidity between migraine and sleep disorders. This review is timely as the topic is relevant and the review explores both clinical studies suggesting the association as well as studies trying to investigated the neurobiological bases of this comorbidity. The article is clear and well written.
I only have a few comments to improve some aspects.
- While it's true that in the majority of cases migraine involves a unilateral pain, bilateral pain can also occur and this might be reported in the paragraph at page 1, lines 27 - 37
- It is not clear whether the authors aimed to conduct a narrative or systematic review and this should be specified. While the authors report some of the criteria used to conduct the search, they do not specify how many articles were retrieved and excluded, or based on which reasons
- Tables are informative but I think they would look better using left alignment for the text. Also, in Table 2 there are additional blank spaces between some of the words
- I didn't find a Table 1, so I think that Table 2 should be renamed in Table 1 and all the following should be renamed accordingly
Author Response
Dear Reviewer,
Thank you very much for your review. All comments and suggestions from the reviewers have been included in the manuscript. English language and style was check by native speaker. The centering of the text in the tables was due to the publisher's sceptical pattern, similar to the manuscript. In the original version it is left-aligned.
Hopefully, the revised version of the manuscript, considering the above issues, would be found suitable for publication.
Authors